# The Use of Neurons Derived from Pluripotent Stem Cells to Study Nerve–Cancer Cell Interactions

**DOI:** 10.3390/ijms26073057

**Published:** 2025-03-27

**Authors:** Adriana Jiménez, Adolfo López-Ornelas, Neptali Gutiérrez-de la Cruz, Jonathan Puente-Rivera, Rodolfo David Mayen-Quinto, Anahí Sánchez-Monciváis, Iván Ignacio-Mejía, Exsal M. Albores-Méndez, Marco Antonio Vargas-Hernández, Enrique Estudillo

**Affiliations:** 1División de Investigación, Hospital Juárez de México, Mexico City 07760, Mexico; adriana.jimenez@salud.gob.mx (A.J.); adolfo.lopez@salud.gob.mx (A.L.-O.); jopuenter@hotmail.com (J.P.-R.); 2Hospital Nacional Homeopático, Hospitales Federales de Referencia, Mexico City 06800, Mexico; 3Escuela Militar de Graduados de Sanidad, Secretaría de la Defensa Nacional, Batalla de Celaya 202, Lomas de Sotelo, Miguel Hidalgo, Ciudad de México 11200, Mexico; gutierrezdlc911@gmail.com (N.G.-d.l.C.); david.invest.emgs@gmail.com (R.D.M.-Q.); amoncivais11@gmail.com (A.S.-M.); ivanignacio402@gmail.com (I.I.-M.); albores_09@hotmail.com (E.M.A.-M.); mavh78@yahoo.com.mx (M.A.V.-H.); 4Laboratorio de Reprogramación Celular, Instituto Nacional de Neurología y Neurocirugía Manuel Velasco Suárez, Mexico City 14269, Mexico

**Keywords:** neurogenesis, sympathetic neurons, axonal growth, tumor microenvironment, axonogenesis, tumor on a chip

## Abstract

Tumor innervation is a complex interaction between nerves and cancer cells that consists of axons invading tumors, and its complexity remains largely unknown in humans. Although some retrospective studies have provided important insights into the relationship between nerves and tumors, further knowledge is required about this biological process. Animal experiments have elucidated several molecular and cellular mechanisms of tumor innervation; however, no experimental models currently exist to study interactions between human cancer and nerve cells. Human pluripotent stem cells can differentiate into neurons for research purposes; however, the use of these neurons to study interactions with cancer cells remains largely unexplored. Hence, here we analyze the potential of human pluripotent stem cells to study the interaction of cancer cells and neurons derived from human pluripotent stem cells to unravel the poorly understood mechanisms of human tumor innervation.

## 1. Introduction

Tumor innervation has recently gained attention since multiple studies have demonstrated its active role in cancer progression toward a more aggressive phenotype [1,2,3]. 

The tumor microenvironment has traditionally been described as a complex interplay of immune cells, fibroblasts, and the extracellular matrix. However, recent research highlights the nervous system as an integral component of cancer biology, with significant implications for tumor progression and therapy [4,5,6]. 

Rather than being passive components of the tumor microenvironment, peripheral nerves actively modify tumor progression in prostate, colorectal, stomach, skin, and breast cancers [3,7,8,9,10,11,12]; the interactions between cancer cells and nerve cells promote neurobiological processes, such as the generation of new neurons from neural stem cells [13,14]. This process is termed neurogenesis and normally occurs during the embryonic development and postnatal life of mammals, mainly in the subventricular and ventricular zones of the lateral ventricles, and in the dentate gyrus of the hippocampus. However, the abundance and period through which human postnatal neurogenesis persists is still under debate [15,16,17,18]. Among other mechanisms that involve increased innervation in tumors is axonogenesis [7,8], which is the formation of an axon from a soma, and it requires the orchestration of multiple genes and splicing processes to undergo neuronal polarization, neurite outgrowth, and axon specification so a correct axon morphogenesis can be achieved [19,20]. Finally, the neural reprogramming of sensory neurons to adrenergic ones is also a cellular mechanism that contributes to tumor innervation [9].

A growing body of evidence highlights the critical role of neuronal interactions in tumor progression, therapy resistance, and metastasis. While conventional two-dimensional (2D) cultures offer simplicity and accessibility, they often fail to capture the spatial complexity and cell diversity required to study perineural invasion (PNI) and neural recruitment. In contrast, organoid-based systems, including those derived from embryonic, induced pluripotent, or patient-derived stem cells, have emerged as powerful platforms that recapitulate in vivo architecture, microenvironmental cues, and tumor heterogeneity.

Stem cells are endowed with the capacity for both self-renewal and specialization, making them essential for tissue maintenance, regeneration, and development. They are traditionally categorized as either embryonic stem cells (ESCs), which originate from the inner cell mass of the blastocyst, or adult stem cells, which are found in various tissues, such as the bone marrow or the intestinal epithelium [21]. Although adult stem cells can differentiate into fewer cell types (often confined to a specific tissue or lineage), ESCs are considered pluripotent, meaning that they can give rise to cells of all three germ layers (ectoderm, mesoderm, and endoderm) [22].

A remarkable breakthrough in this field was the discovery of induced pluripotent stem cells (iPSCs), generated by reprogramming adult somatic cells, such as fibroblasts, back to a pluripotent state [23]. This process avoids several ethical and logistical hurdles tied to ESCs and provides a patient-specific platform for research. Critically, iPSCs preserve pluripotent features, allowing scientists to guide them toward multiple lineages, including diverse types of neurons, under controlled conditions [24]. By creating iPSC-based models that integrate neuronal cells with tumor cells, researchers can conduct nuanced investigations into the ways nerves might drive or modulate cancer progression. Such systems offer a closer approximation of human biology than conventional cell lines do, paving the way for fresh insights into neural–tumor interactions and, potentially, novel avenues for therapeutic intervention.

A complementary strategy to iPSC-based approaches involves organoid cultures, which are three-dimensional, stem cell-derived structures that mirror key histological and functional traits of native tissues [25]. In oncology, tumor organoids capture the cellular diversity and architecture of original tumors more faithfully than traditional monolayers do while also permitting co-culture with iPSC-derived neurons [26]. This setup allows the dissection of neuron–tumor interactions in an environment that better reflects in vivo conditions, thereby shedding light on the mechanisms of cancer innervation and revealing potential therapeutic targets. 

In this review, we describe how tumor innervation modulates the properties of cancer cells and how human pluripotent stem cells can enhance our understanding of this biological process. We discuss key studies demonstrating how stem cell-based organoids and their 2D counterparts increase our understanding of tumor innervation across multiple cancer types.

## 2. Innervation of Different Types of Cancer

### 2.1. Breast Cancer

The relationship between breast cancer and peripheral nerves defines tumor behavior. Clinical data from breast cancer samples revealed a correlation between nerve thickness and malignancy, as thicker nerves were associated with lymph node metastasis and shorter disease-free survival [27].

Moreover, preclinical models demonstrated that vagus nerve activity has a regulatory function on breast cancer since its activation appears to inhibit metastasis [28]. Further studies showed a crosstalk between peripheral nerves and breast cancer. Nerve ablation revealed that breast tumor innervation contributes to tumor growth and promotes a proinflammatory microenvironment by inducing interleukin-6 (IL-6) production and macrophage recruitment [29]. Preclinical results have helped to elucidate tumor innervation mechanisms, demonstrating that vascular endothelial growth factor (VEGF) plays a key role in breast cancer innervation by promoting neurite outgrowth. Remarkably, blocking VEGF receptor 1 (VEGFR1) or its downstream effector, actin-related protein 2/3 complex (ARP2/3), ablates tumor innervation. These data suggest that VEGF signaling elements are a potential therapeutic target to counteract breast cancer progression [30]. 

Adrenergic nerves are known to positively regulate cancer cell proliferation through the activation of β-adrenergic receptors, which also mediate nerve growth factor (NGF) production and secretion, thereby promoting tumor innervation [31]. Additionally, the blockade of β-adrenergic receptors with propranolol inhibited the metastasis of triple-negative breast cancer cells, suggesting that blocking sympathetic nerve activity could also contribute to breast cancer treatment [32]. Remarkably, in rodent models, neural stem cells from the subventricular zone migrate to breast cancer cells, differentiate into catecholaminergic neurons, and increase tumor progression [13].

Other neurons that modulate the malignancy of breast cancer are sensory neurons. Direct contact between breast cancer cells and sensory neurons increases the invasive properties and proliferation of breast cancer cells, and PlexinB3 protein has a key role in this process. On the other hand, paracrine signals do not modify the properties of breast cancer cells, thus suggesting that direct contact between malignant cells and sensory nerves is an important factor in breast cancer progression [33].

### 2.2. Lung Cancer

The lungs receive dense innervation from the peripheral nervous system, including sympathetic innervation from thoracic nerves T2 to T7 and parasympathetic innervation from the vagus nerve. Additionally, the lungs receive innervation from sensory neurons [6]. This information suggests that lung innervation could be a key element in the modulation of tumor progression.

Lung adenocarcinoma (LADC) is the most common subtype of non-small-cell lung cancer (NSCLC), which represents one of the most frequent causes of cancer-related death worldwide. The evaluation of autonomic nerves in LADC specimens showed increased nerve density in samples derived from high-risk LADC patients. While sympathetic fibers were found in paratumoral areas, parasympathetic fibers were observed inside the tumor. Moreover, the high density of nerves from both branches of the autonomic system was correlated with poor prognosis and recurrence-free survival in patients not treated with adjuvant therapy [34]. Data from multiple studies suggest that the increase in the β-adrenergic receptor correlates with a poor prognosis; however, there is still a lack of consistent evidence to support the effectiveness of beta-blockers as a treatment for lung cancer [35]. In other work, the chemical denervation of sympathetic nerves in a mouse model of small-cell lung cancer (SCLC) reduced tumor growth; furthermore, the sympathetic effect on tumor growth was mediated by the adrenergic receptor ADRB2 since its inhibition reduced the growth of SCLC xenografts and human organoids by disrupting PKA signaling [36].

On the other hand, cholinergic nerves invade lung tumors in mice, strongly suggesting that they modulate their progression since lung cancer cells express nicotinic receptors. Indeed, lung tumors are close to cholinergic innervation and have an increase in calcium signaling, indicating that tumoral cells respond to the cholinergic activity. Remarkably, the induction of electrical activity in the neuroendocrine cell subpopulation of lung cancer cells increases their colony formation capacity, thus suggesting that electrical activity positively modulates tumor aggressiveness. In line with this evidence, the inhibition of electrical activity increases the survival time of mice bearing lung cancer tumors [37]. This novel information further supports that adrenergic and cholinergic signaling coexist in the microenvironment of lung cancer.

There is robust evidence that lung cancer cells also modulate the recruitment of peripheral nerves. The co-culture of the human NSCLC cell lines A549 and H460 with PC12 cells from rat pheochromocytoma induced the growth of neurites in PC12 cells, an effect that was not observed when PC12 cells were co-cultured with non-cancerous lung cells. In addition, high expression levels of the neurotrophic factors NGF, brain-derived neurotrophic factor (BDNF), neurotrophin (NT)-3, and NT-4 were observed in both NSCLC cell lines, and the injection of NGF in mice with lung tumors increased the size of the tumors and the expression of the pan-neuronal marker PGP9.5, Ki-67, and N-cadherin. Moreover, tumors treated with NGF displayed increased levels of serotonin, which promoted proliferation, colony formation, and invasion in NSCLC cells in a mechanism partially dependent on the 5-hydroxytryptamine receptor 1D (HTR1D) [38]. Interestingly, SCLC is one of the most metastatic cancers. It was demonstrated that SCLC cells express neuroendocrine markers, but metastatic SCLC also expresses neuronal markers, such as neuron-specific enolase (NSE), which correlated with poor survival. In addition, in a mouse model, metastatic SCLC can induce the growth of axon-like protrusions through the expression of genes that regulate axonogenesis, axon guidance, and neuroblast migration, such as Gap43 and Fez1, whose knockdown reduced the metastatic capacity of SCLC xenografts [39].

Finally, other non-neuronal cells of the nervous system, such as Schwann cells, also contribute to lung cancer progression. Schwann cells are glial components of the peripheral nervous system that support and regulate the function of the peripheral neurons. Schwann cells supply trophic factors needed for nerve regeneration and can induce local immune reactions; the interaction of these cells with the resident macrophages in the nerves has been proposed to be responsible for cancer-induced pain. The presence of Schwann cells in lung tumors has been demonstrated in animal models and samples from patients, which correlated with worse prognosis because advanced stages displayed the highest levels of these cells. In animal models, it was observed that Schwann cells participate in the epithelial–mesenchymal transition (EMT) by activating the phosphoinositide 3-kinase (PI3K)/protein kinase B (AKT)/glycogen synthase kinase-3 beta (GSK-3β)/Snail-Twist signaling pathway, hence supporting the invasiveness and metastasis of lung cancer [40].

### 2.3. Prostate Cancer

Prostate cancer is the second most common neoplasia in men worldwide, with nearly 1.2 million new cases and 400,000 deaths in 2022 [41]. The relationship between prostate cancer cells and nerves has been mainly described in PNI, which is a mechanism for the spreading of tumor cells. Interestingly, a study that compared the proliferative and apoptotic index in prostate cancer specimens showed that carcinoma with PNI displayed lower apoptotic bodies than cancer without PNI [42]. 

Tumor innervation may participate in cancer development since prostatic preneoplastic lesions display an increase in nerve density. Furthermore, ganglion size and number of neurons in these ganglions are higher in cancerous prostate tissue samples than in non-cancerous prostate tissue [8]. Additionally, the presence of autonomic nerves in human prostate cancer correlates with the clinical outcome [43], suggesting a communication between nerves and cancer cells to promote invasion. These findings align with the analysis of 98 patient samples with and without PNI, which revealed the presence of intraprostatic nerves containing sympathetic and parasympathetic fibers in both sample types [44]. The presence of autonomic nerves was also found in other studies, but their density was lower in prostate cancer compared to benign hyperplasia and normal prostate; however, nerve density, mainly sympathetic, increased in the periphery of the tumors [45,46]. 

Prostate tumors induced by the inoculation of human prostate cancer cell line PC-3 in immunosuppressed mice developed infiltrating sympathetic and parasympathetic fibers that emerged from the normal prostatic tissue. In this context, parasympathetic stimulation with a non-selective muscarinic receptor agonist, carbachol, increased tumor dissemination in a mouse model of prostate cancer [43]. In the same way, the ablation of adrenergic nerves or the knockout of β2- and β3-adrenergic receptors in mice significantly reduced tumor growth and metastasis. Moreover, the deletion of β2 receptors in the endothelial cells of mice reduced the progression of prostate cancer by altering cell proliferation and the oxidative metabolism of endothelial cells [47]. α-adrenergic receptors have also been involved in prostate cancer; the RNA sequencing from prostate tumor samples and non-cancerous tissue of 65 patients showed that α-adrenoreceptors were the main overexpressed transcript in tumor specimens [48].

The treatment of hormone-sensitive prostate cancer with androgen deprivation therapy (ADT) subsequently leads to castration-resistant prostate cancer, which can undergo neuroendocrine differentiation, a process that could be dependent on β2-adrenergic receptors [49]. The contribution of sympathetic innervation in the neuroendocrine transformation of prostate cancer cells was also supported by the norepinephrine-induced expression of neuroendocrine markers in human cell lines of prostate cancer and the antagonism of β receptors, which inhibits the neuroendocrine progression in animal models [50]. Overall, this information strongly supports a modulatory effect of sympathetic and parasympathetic innervation in prostate cancer progression.

Mechanisms involved in the development of resistant prostate cancer also include the release of growth factors and chemokines [51]. NGF is one of the most studied neurotropic factors in the growth and survival of sensory and sympathetic nerves [52,53]. Crosstalk has been described between the androgen receptor and NGF in human prostate cancer cells, observing that the inhibition of androgen signaling impaired NGF-induced cell proliferation, and vice versa; the blockade of the NGF receptor, tropomyosin receptor kinase A (TrkA), decreases the mitogenic effect of androgen [54]. NGF can interact with two receptors, TrkA and NGFR, which were found in several types of cancer, including prostate cancer; they both play a role in prostate cancer progression. It has been proposed that TrkA has oncogenic effects, while NGFR is considered to have oncosuppressive activity. However, these findings are controversial. In addition, it was found that NGF participates in the neuroendocrine differentiation of prostate cancer since the inhibition or knockdown of NGF prevents neuroendocrine differentiation through a mechanism dependent on the cholinergic muscarinic receptor 4 (CHRM4) [51]. Nevertheless, NGF can induce the reinnervation of perivascular nerves in prostate tumors in mice, leading to the suppression of tumor growth, possibly through the regulation of vascular tone and blood flow to the tumor [55].

### 2.4. Pancreatic Cancer

Pancreatic ductal adenocarcinoma (PDAC) is one of the most aggressive types of cancer; notably, PDAC is associated with high rates of PNI [56]. The innervation of the tumor microenvironment in PDAC has been shown to promote tumor progression. The presence of nerves by immunodetection with neuronal markers such as S100, protein gene product 9.5 (PGP9.5), and growth-associated protein 43 (GAP-43) was demonstrated in pancreatic cancer (PC) samples, in which larger nerves were associated with worse survival [57].

The peripheral nervous system is implicated in PDAC. The role of sensory and autonomic innervation in this cancer has been primarily studied in animal models. Using an autochthonous model of PDAC in mice, it was observed that cancerous cells were able to migrate along the sensory nerves, and the ablation of these nerves not only prevented the PNI but also delayed the formation of pancreatic intraepithelial neoplasia and improved survival [58]. Moreover, sensory innervation is known to have a role in the initiation and maintenance of inflammation in the exocrine pancreas and has been related to pathological processes such as pancreatitis [59]. Increased nerve density was found in pancreatic intraepithelial neoplasia in mice, in which a subpopulation of neuroendocrine cells was observed that express the receptor for substance P (SP), Neurokinin 1-R. Interestingly, sensory neurons promoted the proliferation of pancreatic organoid cultures in a mechanism dependent on SP/Neurokinin 1-R signaling [56].

Parasympathetic innervation has gained importance in the development of PDAC since vagal innervation promotes the proliferation of healthy exocrine pancreas; for example, vagal hyperactivity can lead to pancreatic hypertrophy, while vagotomy decreases acinar growth. Using a genetically engineered mouse model of PDAC, it was demonstrated that subdiaphragmatic vagotomy accelerated PDAC development, and the administration of bethanechol (muscarinic agonist) suppressed the effects of vagotomy and significantly increased survival in mice with PDAC. The parasympathetic effect that prevented PDAC development was partially associated with CHRM1 and the inhibition of mitogen-activated protein kinase (MAPK)/epidermal growth factor receptor (EGFR) and PI3K/AKT pathways [60]. These findings suggest a protective action of parasympathetic innervation in PDAC.

Systemic stress was associated with accelerated tumor growth in several cancers, including PDAC xenografts. Increased levels of norepinephrine were found in PDAC, which also expresses α- and β-adrenergic receptors; moreover, immunoreactivity to β-adrenergic receptors in peritumoral tissue was associated with poor prognostic factors [60,61]. Furthermore, in a mouse model of PDAC, catecholamines promoted tumor development dependent on the activation of the ADRB2 receptor and NGF secretion, which was reverted by blocking ADRB2 and the NGF receptor [60]. However, another study in mice using 3D imaging of optically cleared tissues showed increased sympathetic innervation in PDAC by the engulfment of the nerves and the emergence of axon terminals in the tumor and found that sympathectomy promotes tumor growth and spread by the increased infiltration of macrophages [62]; this suggests that sympathetic innervation could have an antitumoral action. Although these results are not conclusive, they showed a role for sympathetic innervation in PDAC; further investigation of this branch of the autonomic system must be performed to completely elucidate the effect of sympathetic nerves in PDAC. 

In addition, innervation in PDAC could support the metabolic requirements in the tumor microenvironment. In this context, serine is used in multiple metabolic pathways; hence, it is considered a metabolic substrate relevant for growth in several types of cancer. In exogenous serine-dependent PDAC, serine deprivation induced the expression of NGF to promote innervation and the release of serine by neurons to rescue tumor growth [63]. These results reinforce the role of innervation in cancer to promote cell proliferation and tumor growth. 

A novel strategy has recently been used to identify differential innervation in healthy and cancerous pancreatic tissue. Neurons innervating the organ were retrogradely marked with the fluorescent tracer fast blue; after 5-14 days of tracing, fast blue-positive neurons were isolated using fluorescent activated cell sorting to perform single-cell RNA sequencing (Trace-n-Seq). The healthy pancreas was mainly innervated by noradrenergic neurons and glutamatergic, neurofilament, peptidergic, and non-peptidergic sensory neurons. The total number of neurons per ganglion was similar in healthy tissue and pancreatic tumors; however, dorsal root ganglion sensory neurons were more abundant in PDAC than in non-cancerous pancreas. Through Trace-n-Seq analysis, authors identified a PC nerve expression signature associated with tumor proliferation and relapse [64].

### 2.5. Colorectal Cancer

The gastrointestinal tract is densely innervated by both extrinsic and intrinsic nervous systems. The extrinsic component consists of the sympathetic and parasympathetic branches of the autonomic nervous system, while the intrinsic component is represented by the enteric nervous system (ENS). Often called the “second brain”, the ENS governs gut motility, secretion, and blood flow. Disruptions in these networks are increasingly recognized as contributing factors in various diseases, including cancer [65].

Colorectal cancer (CRC) is one of the leading causes of cancer-related morbidity and mortality worldwide [11,12]. Neural elements can become integrated into the tumor microenvironment, enhancing CRC progression and resistance to therapy. Tumor cells secrete neurotrophic factors, promoting innervation and reinforcing a pro-tumorigenic feedback loop [66]. Additionally, sympathetic and parasympathetic nervous system activity modulates immune responses within the tumor microenvironment, influencing cancer progression. For instance, parasympathetic (vagal) nerves have been shown to promote prostate, gastric, and colorectal cancers, whereas they may suppress tumor growth in breast cancer and PC. In CRC, vagal denervation has been associated with reduced tumor incidence, smaller tumor volume, decreased angiogenesis, and lower expression of cluster of differentiation (CD)31, VEGF, NGF, and β2-adrenergic and M3 receptors (M3Rs) [67].

PNI is a pathological hallmark in which cancer cells infiltrate nerve fibers. In CRC, PNI is linked to increased tumor aggressiveness, recurrence, and poor prognosis. This phenomenon involves complex tumor–neural interactions, including the secretion of neurotrophic factors and the upregulation of adhesion molecules that facilitate nerve infiltration. A better understanding of the molecular mechanisms underlying PNI could lead to more effective therapeutic strategies [68].

CRC cells also express muscarinic acetylcholine receptors, particularly M3Rs, which are G protein-coupled receptors involved in smooth muscle contraction, glandular secretion, and neuronal signaling [69]. Acetylcholine, produced by both neural elements and tumor cells, activates M3R to drive tumor growth and metastasis. The inhibition of M3R significantly reduces tumor progression, highlighting its role in CRC pathogenesis [70,71]. Modulating neural pathways, including β-adrenergic and muscarinic receptors, presents a promising approach to influencing immune responses in CRC [72].

Netrin-1, a ligand for the deleted in colorectal cancer (DCC) receptor, plays a multifaceted role in CRC. It promotes neuronal sprouting into the tumor microenvironment via pathways involving focal adhesion kinase (FAK) and Rho GTPases (Rac1, Cdc42), enhancing tumor growth and potentially contributing to cancer-related pain [73]. The role of Netrin-1 is context-dependent: it fosters tumor progression in chronic inflammation through DCC pathway upregulation [74] but has also been implicated in glioma cell motility inhibition [75]. Furthermore, Netrin-1 supports cancer cell stemness, counteracted by bone morphogenetic protein (BMP) signaling, suggesting a dynamic interplay between these pathways in shaping tumor behavior [76,77]. Given DCC’s tumor-suppressive function, blocking Netrin-1/DCC interactions has emerged as a potential therapeutic approach to disrupting prosurvival signaling in cancer cells [78]. Additionally, serum Netrin-1 has been identified as a potential CRC biomarker [79], reinforcing its clinical relevance.

Moreover, it was demonstrated that enteric serotonergic neurons critically regulate colorectal cancer stem cell (CSC) self-renewal and tumorigenesis through serotonin (5-HT) signaling. Colorectal CSCs selectively overexpress the 5-HT receptors HTR1B, HTR1D, and HTR1F, which activate Wingless-related Integration Site (Wnt)/β-catenin signaling, a key pathway sustaining CSC pluripotency. Gut microbiota-derived metabolites, such as isovalerate, amplify neuronal–tumor crosstalk by upregulating tryptophan hydroxylase 2 (Tph2), increasing 5-HT synthesis. The pharmacological inhibition of 5-HT via p-chlorophenylalanine (pCPA) reduced CSC populations and the tumor burden in murine models, whereas HTR1B/1D/1F blockade suppressed liver metastasis, underscoring the role of serotonin in CSC-driven progression. This observation challenges conventional views of enteric neurotransmission, revealing that serotonergic dominance is involved in CSC regulation. To dissect this niche, the authors advocate for neural-integrated organoid models that replicate the tumor microenvironment, enabling the precise study of neurotransmitter effects on CSC behavior. Quantitative analyses linked elevated 5-HT levels to enriched Wnt targets (c-Myc, Cyclin D1) and enhanced sphere formation, positioning serotonin as both a prognostic biomarker and a therapeutic target [80].

## 3. The Use of Pluripotent Stem Cells to Study Cancer Biology

Altogether, these findings strongly suggest that targeting cancer innervation could be a promising therapeutic strategy. However, there is a lack of knowledge regarding the mechanisms that underlie human tumor innervation [81]. Most of the studies are limited to retrospective research of clinical data and animal models, as mentioned above. This is partially due to the limited platforms available to study cancer innervation in a context where only human cancer cells and human nerves could be evaluated.

Human embryonic stem cells (hESCs) are derived from the inner mass of human blastocysts [82], while human induced pluripotent stem cells (hiPSCs) can be obtained from blood cells or dermal fibroblasts of any individual through cell reprogramming [83,84,85]. Both cell types are pluripotent stem cells (PSCs) and give rise to virtually every cell type from the three germ layers; therefore, this property can be harnessed to obtain human neurons derived from well-established hESCs or hiPSCs with minimal or nonexistent bioethical concerns (Figure 1) [86]. Nevertheless, these study models have not been exploited sufficiently, and their potential to investigate human tumor innervation is still far from being reached [4].

### The Use of PSCs with Microfluidic Devices in Neurobiology and Cancer Biology

Microfluidic platforms are used to study the connectivity patterns of several neuronal types since some microfluidic designs allow the compartmentalization of two neuronal populations in order to establish communication between them by exclusively using their axons or dendrites, which pass through microgrooves that only allow the entrance of axonal or dendritic processes [87]. Furthermore, these microfluidic devices also allow the evaluation of potential biomolecules to induce axonal growth by plating neurons and biomolecules in different compartments. Recent studies have tested this experimental strategy by evaluating the potential of semaphorin 3C and creatine to promote the axonal growth of dopaminergic neurons and motor neurons, respectively [88,89]. Remarkably, microfluidic devices are also useful for studying the interactions between axons of motor neurons and muscle cells in order to investigate the alterations of the neuromuscular junction in a neurodegenerative context, such as amyotrophic lateral sclerosis [90]. 

Microfluidic devices have allowed the study of the tumor microenvironment in a more dynamic context. The diversity of microfluidic devices provides versatile platforms that can be customized to study several biological processes. This study model has shed light on previously unknown aspects of tumor cell biology, such as angiogenesis, cell migration, and the diffusion of nutrients among malignant cells [91,92]. Nevertheless, this technology has not been completely exploited to study important emerging processes such as tumor innervation. However, microfluidic models for studying neuromuscular junctions strongly suggest the high viability of studying axonal growth toward cancer cells in a similar design, and information can be obtained regarding tumor innervation.

Recent studies have shed light on tumor innervation using microfluidic devices with axonal growth chambers, enabling the retrograde labeling of rodent sensory neurons by applying wheat germ agglutinin (WGA) and cholera toxin subunit B (CTB) dyes in the breast cancer cell compartment. WGA and CTB were able to move through the axons in the microgrooves and reach sensory neurons in the other compartment; however, breast cancer cells accumulated a considerable amount of these dyes [93]. This pioneering information shows the potential use of these axonal chambers to study human tumor innervation processes in a more controlled microenvironment with the use of human neurons. 

One particular drawback in studying human nerves and cancer cells is the limited availability of human neurons. Nonetheless, human pluripotent stem cells (hPSCs) or hiPSCs can overcome this issue. Co-culturing cancer cells with human adrenergic neurons in microfluidic platforms could provide novel insights into this process (Figure 2). Since adrenergic neurons have been widely associated with tumor innervation and progression, and it is viable to yield adrenergic neurons from either hPSCs or hiPSCs through protocols already established [94]. hPSC-derived adrenergic neurons could be further genetically modified in order to express a reporter gene, such as the enhanced green fluorescent protein, to perform time-lapse microscopy and follow the innervation process. 

Using mitoTracker, mitochondrial behavior in axons can also be monitored. This experimental approach has previously been used to analyze the movement patterns of mitochondria that reside in dopaminergic neurons derived from Parkinson’s disease patients [87] and could provide information on how mitochondria contribute to axonal growth toward cancer cells.

The use of microfluidic platforms could also be useful for evaluating morphological changes, such as the EMT in cancer cells, and elucidating how this interaction modifies the cell biology of both neurons and cancer cells in a humanized context. Finally, this strategy is useful for testing potential drugs to inhibit tumor innervation since this biological process has recently emerged as a potential therapeutic target (Figure 1).

## 4. The Use of Tumor Organoids to Study Cancer Innervation

The rapidly evolving field of cancer neuroscience delineates how tumors exploit neural circuits through electrochemical synapses, paracrine signaling, and neurotransmitter release to drive progression. Specifically, the PNI is a prognostic marker that is prevalent in 80% of pancreatic, 40–50% of prostate, and 30% of breast cancers, with PNI-positive cases correlating with significantly poorer outcomes. In glioblastoma (GBM), glutamatergic neuron-to-tumor synapses increase proliferation, whereas optogenetic neuronal activation increases invasion. β-adrenergic signaling has emerged as a key mediator, where stress-induced sympathetic activity accelerates metastasis, and β-blockers reduce tumor growth in preclinical models. Neurotransmitters (acetylcholine, norepinephrine, and serotonin) modulate angiogenesis, immune evasion, and survival, underscoring their dual role as metabolic regulators and oncogenic signals. The integration of neural components into organoid models to dissect tumor–nerve crosstalk and screen therapies targeting neurotrophic factors (e.g., NGF, BDNF) or glutamate receptors (α-Amino-3-Hydroxy-5-Methyl-4-Isoxazole Propionic Acid (AMPA)/N-Methyl-D-Aspartate (NMDA)) is crucial. This paradigm turns neural circuit disruption into a strategic frontier in precision oncology, particularly for cancers leveraging neural plasticity for therapy resistance [95].

### 4.1. Pancreatic Tumor Organoids and Innervation

It was identified a neuroendocrine-driven mechanism in pancreatic precancerous lesions, demonstrating that sensory neurons amplify pancreatic intraepithelial neoplasia (PanIN) growth via SP–neurokinin 1 receptor (NK1-R) signaling. Co-culturing rodent sensory neurons with rodent PanIN organoids induced increased proliferation, surpassing fibroblast-mediated effects, with SP activating the phosphorylation of the activator of transcription signal transducer and activator of transcription (Stat) 3 in NK1-R+ neuroendocrine cells. In genetically engineered PDAC mice, sensory denervation reduces late-stage PanIN lesions, whereas histopathological analysis reveals increased PanIN-associated nerve density. Pharmacological NK1-R or Stat3 inhibition attenuated organoid growth, confirming the targetability of neuro-epithelial crosstalk. This study established PanIN neuroendocrine cells as mediators of neural recruitment, with organoid–nerve co-cultures revealing tumor cell axonotropism. These findings underscore the utility of organoid models for dissecting early neural–tumor interactions, suggesting that NK1–R/Stat3 blockade is a strategy to disrupt premalignant progression. This work further highlights the potential of using innervated organoids to screen neuromodulatory therapies, particularly for cancers such as PDAC, where perineural invasion precedes metastasis [56].

In addition, it was established a murine co-culture model integrating PC organoids from *Kras^G12D/+*; *Trp53^R172H/+*; *Pdx1-Cre* (KPC) mice with hiPSC-derived neural crest cells or dorsal root ganglion (DRG) explants to investigate neuropathic mechanisms in PDAC. Compared with wild-type controls, tumor organoids presented increased βIII-Tubulin+ neural density after 7 days, accompanied by glial fibrillary acidic protein (GFAP)+ glial cell recruitment, reflecting early neuro-glial activation. Transcriptomic profiling revealed that KPC organoids upregulated neurotrophic factors (NGF and glial cell line-derived neurotrophic factor (GDNF)) and axon guidance molecules (semaphorin 3A and Ephrin type A receptor 4 (EPHA 4)), implicating tumor-secreted signals in neural remodeling. Functional validation revealed that anti-NGF treatment reduced neurite outgrowth, linking NGF signaling to cancer-driven innervation. This platform demonstrates that PDAC organoids actively recruit and remodel neuronal networks, providing a physiologically relevant ex vivo system for dissecting PNI mechanisms. The model’s adaptability to human patient-derived organoids (PDOs) and high-throughput screening make it a strategic tool for evaluating neuro-targeted therapies (e.g., tropomyosin receptor kinase inhibitors) and precision oncology approaches aimed at disrupting tumor–nerve crosstalk in PDAC progression [96].

Recent methodological advances address critical technical barriers in visualizing neural–tumor interactions within three-dimensional (3D) pancreatic organoid models. Their optimized whole-mount immunofluorescence protocol for extracellular matrix (ECM) gel-embedded organoids resolves the challenges of antibody penetration and autofluorescence through fructose–glycerol clearing and a brief 2% paraformaldehyde fixation, preserving structural integrity while enhancing signal clarity. After 7 days of co-culture, confocal imaging confirmed the directional growth of βIII-Tubulin+ neurons toward E-Cadherin+ pancreatic organoids with GFAP+ glial cell colocalization, features absent in noninnervated controls, which validated the methodological specificity. Quantitative analysis revealed that 100% of the innervated organoids were ≤3 mm in size and that the perimeter exhibited complete antibody penetration versus 0% in their larger counterparts, establishing size thresholds for reliable imaging. Compared with standard protocols, extended antibody incubation (72 h at 4 °C) improved the intraorganoid labeling efficiency by 40%. This platform enables precise 3D mapping of tumor–nerve interfaces, with implications for identifying neurotrophic signaling pathways (e.g., GDNF/Rearranged during Transfection (RET)) and screening therapeutics targeting neural recruitment in PC models [97].

Interestingly, it was developed a pioneering co-culture model combining human brain organoids (hBrOs) and murine PC organoids (mPCOs) to recapitulate the neuroinvasion dynamics of PC. Within 7–11 days, mPCOs selectively invade hBrOs, triggering neural projection retraction, a 30–40% increase in neuronal apoptosis, and upregulation of the neuroinflammatory markers NOD-, LRR-, and Pyrin Domain-Containing Protein 3 (NLRP3) and IL-8, mirroring the hallmarks of clinical PNI. mPCOs secrete the neurotrophic factors GDNF and BDNF, which facilitate structural remodeling akin to the in vivo spinal cord infiltration observed in murine models. Proliferative Ki-67+ neuronal progenitor cells were concentrated at invasion margins, suggesting compensatory neurogenesis during cancer encroachment. The pathophysiological importance of the model is underscored by its ability to replicate the high PNI incidence of PC (80–100% in patients), providing a platform for dissecting nerve–tumor crosstalk. By demonstrating WNT/β-catenin-driven invasion pathways and neuroinflammatory cascades, this system enables targeted interrogation of the mechanisms underlying neural remodeling. Its adaptability to PDOs positions it for personalized drug screening, particularly for therapies that block neurotrophic signaling (e.g., GDNF/RET inhibitors) or mitigate cancer-induced neuroinflammation. These advances bridge critical gaps in the study of the role of PNI in metastasis and pain, suggesting translational potential for therapies aimed at disrupting neural recruitment in PC [98].

### 4.2. Colon Tumor Organoids and Innervation

Recently it was elucidated a neuronal–epithelial axis wherein vasoactive intestinal peptide (VIP)-producing enteric neurons regulate intestinal homeostasis by modulating epithelial fucosylation via Erk1/2–c-Fos-dependent fucosyltransferase 2 (Fut2) activation. Co-culture experiments with mice enteric neurons and mice intestinal organoids demonstrated that VIP treatment upregulated Fut2 expression, enhancing the α1,2-fucosylation critical for beneficial Bifidobacteria colonization. The disruption of VIP signaling (via vagotomy or receptor blockade) reduces epithelial fucosylation and decreases *Bifidobacteria* abundance while increasing pathogenic *Enterococcus faecalis* colonization by 70% in fucosylation-deficient mice, which is correlated with a greater risk of alcohol-associated liver disease. These findings implicate VIPergic neurons as key regulators of mucosal–microbial crosstalk, with broader implications for cancer innervation. Given the established role of dysbiosis in colorectal and pancreatic cancers, this study suggests that the neurogenic control of epithelial glycosylation may influence tumorigenesis by shaping the microbiota composition and epithelial barrier integrity. Organoid–neuron co-culture models, as demonstrated here, provide a platform to dissect how neurotrophic signals (e.g., VIPs) modulate CSC niches or tumor-associated microbiota, potentially informing therapies targeting neural–microbial axes in gastrointestinal cancers [99].

In addition, α-synuclein (α-syn) accumulation in the enteric nerves of pancreatic/colorectal tumors was observed. Interestingly, the study of a gut-to-vagus nerve axis for pathogenic α-syn propagation using co-cultures of human α-syn-expressing intestinal organoids with α-syn-deficient nodose ganglia neurons demonstrated intercellular transfer within 5 days, with neurons accumulating intracellular aggregates. In A53T α-syn transgenic mice, quantitative assays revealed fibril-templating activity in the duodenum, vagus nerve, and dorsal motor nucleus, whereas subdiaphragmatic vagotomy abolished hindbrain transmission. Real-time quaking-induced conversion (RT-QuIC) assays confirmed that α-syn seeding in the nodose ganglia by 3 months postinduction was absent in vagotomized mice. Using SNCAbow Cre-loxP models, a study revealed that gut-derived mutant α-syn (A30P, A53T) traffics to afferent neurons, revealing that α-syn+ enteroendocrine cells directly contact vagal neurites before aggregate internalization. These findings mechanistically link gut mucosal α-syn to neuroinvasion, with implications for cancer innervation. The organoid–neuron co-culture system provides a framework to explore whether tumor cells similarly transfer oncogenic or neurotrophic cargo (e.g., exosomes, signaling proteins) to recruited nerves. This work positions gut-innervating neurons as dynamic participants in disease progression, suggesting that therapeutic strategies targeting intercellular transfer mechanisms may disrupt neurotrophic cancer processes [100].

### 4.3. Lung Tumor Organoids to Study Innervation

Hypoxia drives CSC-mediated invasion in lung tumor organoids (LTOs) since 1% O₂ increased collective migration compared with normoxic controls, alongside hypoxia-inducible factor 1-alpha (HIF-1α) upregulation, VEGF induction, and EMT activation through zinc finger e-box binding homeobox 1 (ZEB1) and Snail family transcriptional repressor 1 (SNAIL1). Single-cell RNA-seq revealed a hypoxia-enriched CSC subpopulation with elevated Wnt/Notch signaling, which displayed greater clonogenicity than bulk tumor cells. Neuronal co-cultures revealed bidirectional crosstalk, with LTOs migrating toward SH-SY5Y neuroblastoma spheroids and inducing neurite outgrowth in 60% of the cases. These findings implicate hypoxia as a driver of CSC plasticity and neural recruitment, which are relevant to the PNI observed in SCLC. The model suggests LTOs as tools to screen HIF-1α inhibitors or neurotrophic pathway blockers, offering strategies to disrupt hypoxia-fueled neural–tumor crosstalk in precision oncology [101].

### 4.4. Prostate Tumor Organoids and Innervation

Paradigmatically, it was revealed a mechanism in cancer neuroscience, demonstrating that central nervous system (CNS)-derived doublecortin-positive (DCX+) neural progenitors migrate from the subventricular zone (SVZ) to prostate tumors, driving the de novo neurogenesis critical for tumor progression. In murine models, DCX+ progenitors infiltrate tumors, where they differentiate into tyrosine hydroxylase-positive adrenergic neurons, and genetic ablation reduces tumor growth and suppresses metastasis. Compared with benign tissues, human prostate cancers are strongly correlated with DCX+ cell density and tumor aggressiveness, with high-risk tumors harboring more DCX+ cells. Tumors recruit these progenitors by disrupting the SVZ blood-brain barrier and secreting neurotrophic factors (GDNF, NGF, BDNF), facilitating progenitor homing. This work challenges the traditional view of tumor innervation as solely axon sprouting, proposing instead that cancers hijack CNS neurogenic reservoirs. Targeting this axis via NGF/Trk inhibitors or β-adrenergic blockers could disrupt neurogenesis-driven malignancy. Integrating neural progenitors with PDOs may further elucidate how tumors exploit neurodevelopmental pathways, offering strategies to intercept neural recruitment in prostate, pancreatic, and other neurotrophic cancers [13].

Recent findings revealed that prostate cancer cells exhibit dynamic plasticity under ADT, transitioning from neuroendocrine differentiation to a stem-like phenotype, a process with implications for tumor innervation and therapeutic resistance. Over 90 days of androgen depletion, LNCaP cells upregulated neuroendocrine markers (βIII-tubulin, NSE) followed by stemness factors (CD133, Aldehyde Dehydrogenase 1 Family Member A1 (ALDH1A1), ATP-Binding Cassette Subfamily B Member 1A (ABCB1A)), accompanied by an increase in the pluripotency regulators NANOG and OCT4. This phenotypic shift correlated with chemoresistance, as stem-like cells demonstrated 40–60% reduced sensitivity to docetaxel and androgen receptor antagonists. Central to this plasticity is AMP-activated kinase (AMPK), whose activity decreases during lineage transition, as evidenced by a 70% reduction in phosphorylated AMPK and acetyl-CoA carboxylase. Pharmacological AMPK activation (e.g., capsaicin, A-769662) reversed stem-like traits, restored drug sensitivity, and suppressed EMT markers (VEGF and HIF-1α), while E-cadherin was recovered. These findings underscore the role of AMPK as a gatekeeper of differentiation, with therapeutic potential to counteract neural-like plasticity and perineural invasion mechanisms. Organoid models integrating neuroendocrine or neural co-cultures could elucidate how such plasticity facilitates tumor–nerve crosstalk, whereas AMPK-targeted strategies may disrupt neurotrophic signaling and stemness-driven resistance in prostate cancer [102].

### 4.5. Brain and Glioma Organoids for the Study of Innervation

Preclinical models for diffuse intrinsic pontine glioma (DIPG) by co-culturing tumor cells with human cortical organoids, revealed microenvironment-driven molecular shifts. In mosaic (intermixed tumor–neuron) and assembloid (tumor–organoid juxtaposition) models, the sequential window acquisition of all theoretical mass spectra (SWATH-MS) proteomics revealed that DIPG cells downregulated the expression of extracellular matrix adhesion markers (versican and glypican-1) while increasing the expression of proliferation proteins (Minichromosome Maintenance Complex Component 2 (MCM2) and DNA Polymerase Alpha Subunit 2 (POLA2)), which is indicative of enhanced DNA replication. The mosaic model uniquely activated the neuron-specific BRG1/BRM-associated factor (nBAF) complex, an epigenetic regulator linked to neuronal differentiation, and reduced β1 integrin expression, promoting a migratory phenotype. These findings underscore the role of direct neuron–tumor interactions in driving invasiveness, which are absent in 2D cultures. The assembloid model highlighted metabolic adaptation, suggesting vulnerabilities exploitable by inhibitors. This study advocates the use of PDOs in drug screening, with an emphasis on targeting tumor–microenvironment crosstalk. The future integration of vascularized or region-specific organoids could refine models for DIPG’s stem cell-driven plasticity and improve therapeutic discovery [103].

Remarkably, it was revealed a synaptic hijacking mechanism by which the GBM exploits neural circuits to drive progression, employing rabies virus retrograde tracing to map functional neuron–tumor networks. In patient-derived GBM organoids (GBOs) and brain slices, tumor cells engaged in synaptic-like interactions with glutamatergic and cholinergic neurons, with cholinergic signaling via the CHRM3 accelerating proliferation. The genetic silencing of *CHRM3* reduced tumor growth by 50%, whereas radiotherapy paradoxically amplified neuron–tumor connectivity, a vulnerability exploited by combining radiation with AMPA receptor inhibition, which synergistically suppressed growth. The rabies-mediated Cre-loxP ablation of tumor-connected neurons reduced invasion by disrupting circuit integration, revealing the reliance of the GBM on neural networks for invasion. Longitudinal tracing identified long-range neural circuit cooperation, with tumors recruiting distant neurons to expand their invasive field. These findings put neuron–tumor synapses as actionable targets, suggesting that CHRM3 antagonists or neural activity modulators are adjuvants to standard therapies. This study further establishes GBOs as critical tools for modeling synaptic crosstalk, with implications for developing patient-specific strategies to disrupt neural hijacking. By bridging neurobiology and oncology, this work redefines GBM as a networked disease, offering pathways to dismantle its adaptive circuitry [104].

In line with this, it was established a GBO biobank that faithfully recapitulates tumor heterogeneity, with single-cell RNA sequencing confirming the retention of parental tumor transcriptomic profiles for ≥12 weeks. Xenotransplantation studies demonstrated that 92% of GBOs aggressively infiltrated rodent brains, mirroring human GBM migration along white matter tracts. The platform preserved molecular subtypes, with EGFR-mutant GBOs showing reduced Ki-67+ cell proliferation upon gefitinib treatment, neurofibromin 1-mutant organoids exhibiting trametinib sensitivity, and PI3K-mutant models achieving Ki-67 suppression with everolimus. While the study focused primarily on therapeutic profiling, the model’s preservation of invasive behavior, including neural tract infiltration, positions GBOs as critical tools for dissecting GBM–neuron interactions. The observed infiltration patterns align with clinical evidence of the GBM synaptic hijacking of neural circuits, suggesting future applications in modeling neuron–tumor connectivity. Integrating GBOs with cerebral assembloids could elucidate how tumor cells exploit neuronal activity for progression [105].

Recently it was mapped the synaptic hijacking of neural circuits via the GBM and revealed that tumor cells form functional connections with neurons through glutamatergic (80%) and cholinergic (5%) inputs. Using transsynaptic viral tracing, they revealed that basal forebrain cholinergic neurons synapse onto GBM cells via CHRM3 receptors, inducing calcium oscillations that increase tumor motility and upregulate the promigratory transcription factors FOS and EGR1. The optogenetic activation of cholinergic neurons triggered GBM calcium spikes, whereas CHRM3 blockade reduced migration and prolonged survival in murine models. Cortical regions (S1, M1, and the retrosplenial cortex) provide the densest inputs, highlighting the GBM’s capacity to exploit diverse neurotransmitter systems. These findings suggest neuron–tumor synapses as critical drivers of GBM progression, with CHRM3 inhibition emerging as a therapeutic strategy. This study underscores the ability of GBOs to model synaptic crosstalk, offering a platform for dissecting how neuronal activity influences therapy resistance and immune evasion in neurotrophic malignancies [106].

Collectively, these studies illustrate the ability of organoids, often derived from stem cells, to overcome the limitations of 2D cultures in modeling the complex interplay between tumor cells and the nervous system. By retaining parent–tumor heterogeneity and permitting the incorporation of diverse cell types, organoids enable a deeper exploration of innervation mechanisms, neurotrophic signaling, and perineural invasion (Table 1).

Equally important, 2D models still offer advantages in simplicity, throughput, and mechanistic dissection; however, 3D organoid platforms, including those fused with neuronal or glial components, provide a more physiologically relevant setting for investigating CSCs, therapy resistance, and neural recruitment. Future directions should focus on integrating vascular and immune elements, refining patient-specific approaches, and leveraging CRISPR-based editing to track dynamic evolution, all of which could yield transformative insights for targeting tumor–nerve interactions in precision oncology.

## 5. Conclusions

Cancer treatment requires novel multidisciplinary approaches that consider factors beyond inflammation and vascularization. Tumor innervation has recently gained attention due to its active modulation of cancer behavior in several types of tumors. Therefore, there is a need to develop study models to recapitulate cancer innervation and investigate its role in the progression of tumors. 

Neurons derived from human PSCs hold a promising potential to study the interaction between nerves and tumors that could allow the modeling of tumor innervation. The study of cancer innervation through microfluidic devices and organoids can provide novel insights regarding the cellular processes underlying this phenomenon and could help to identify new therapeutic targets to improve cancer treatment and the clinical outcome of this disease by targeting tumor innervation. 

## Figures and Tables

**Figure 1 ijms-26-03057-f001:**
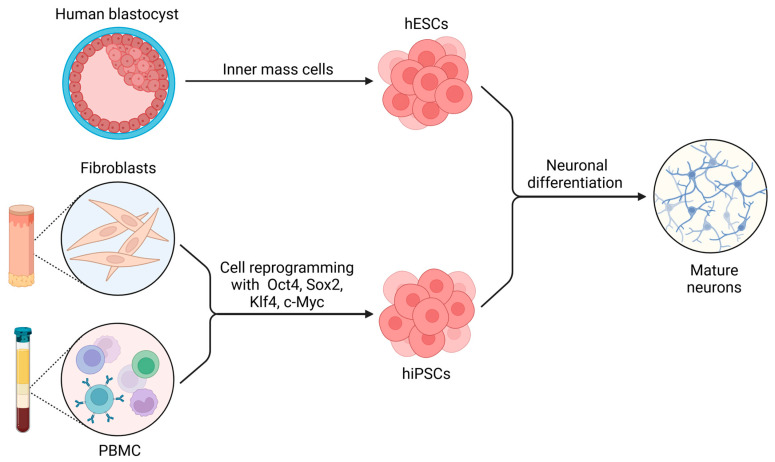
The use of pluripotent stem cells (PSCs) for the generation of neurons. Human embryonic stem cells (hESCs) can be obtained from the inner mass of blastocysts, while human induced PSCs (hiPSCs) can be obtained from the cell reprogramming of fibroblasts or peripheral blood mononuclear cells (PBMCs). Different protocols of differentiation can be used to obtain neurons derived from hESCs and hiPSCs. Figure created with BioRender.

**Figure 2 ijms-26-03057-f002:**
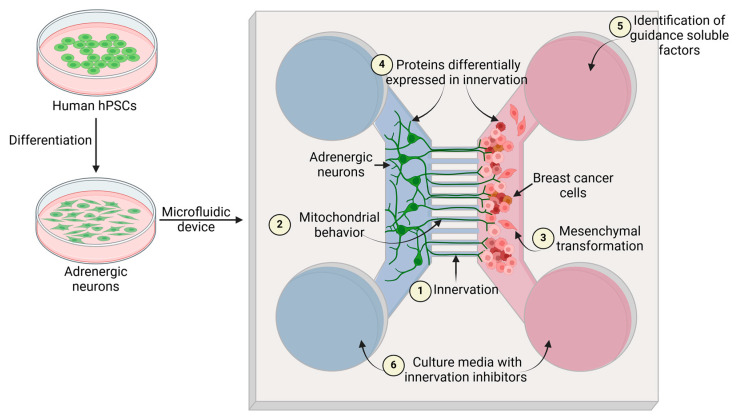
Tumor innervation model using microfluidic devices. Adrenergic neurons widely involved in tumor innervation can be obtained from the differentiation of hPSCs and co-cultured with breast cancer cells in microfluidic devices (1). Furthermore, the following can be evaluated: the mitochondrial behavior in the innervating axons (2), morphological changes mediated by the neuron and tumor interaction such as mesenchymal transformation (3), proteins differentially expressed in neurons and tumor cells that mediate innervation (4), guidance molecules for axons (5), and the tumor progression in the presence of innervation inhibitors (6). Arrows indicate the site where each biological/methodogical process takes place. Figure created with BioRender.

**Table 1 ijms-26-03057-t001:** Models of study that explored tumor innervation using tumor organoids.

Tumor Organoid	Model of Innervation	Effect	Molecular Mechanism	Reference
Pancreas	PanIN organoids with rodent sensory neurons. KPC organoids with hiPSC-derived neural crest cells or DRG explants. Human brain organoids (hBrOs) and murine PC organoids (mPCOs).	PanIN proliferation. Increased βIII-Tubulin+ and GDNF+ cells. Anti-NGF treatment reduces neurite outgrowth. mPCOs selectively invade hBrOs and induce neural projection retraction and apoptosis.	SP/NK1-R/Stat3 signaling. NGF, GDNF, semaphorin 3A, and EPHA4. Neuroinflammatory markers. GDNF and BDNF secretion by mPCOs.	[56] [96] [98]
Lung	LTOs under hypoxia and co-cultured with SH-SY5Y neuroblastoma spheroids.	Hypoxia induces EMT. LTOs migrate toward neuroblastoma spheroids and induce neurite outgrowth.	HIF-1α, VEGF, ZEB1 and SNAIL1. Wnt/Notch signaling.	[101]
Glioma	Co-culture of DIPG with human cortical organoids. Patient-derived GBM organoids (GBOs) and brain slices.	DIPG cells downregulate the expression of ECM adhesion markers and increase the expression of proliferation proteins. Tumor cells engage in synaptic-like interactions with glutamatergic and cholinergic neurons, accelerating proliferation.	Versican, glypican-1 nBAF complex, β1 integrin downregulation. CHRM3	[103] [104]

PanIN: pancreatic intraepithelial neoplasia, KPC: Kras^G12D/+; Trp53^R172H/+; Pdx1-Cre mice, DRG: dorsal root ganglion, LTOs: lung tumor organoids, EMT: epithelial–mesenchymal transition, DIPG: diffuse intrinsic pontine glioma, ECM: extracellular matrix, GBM: glioblastoma.

## Data Availability

Not applicable.

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
