# Peer review of "The Use of Neurons Derived from Pluripotent Stem Cells to Study Nerve–Cancer Cell Interactions"

_ijms, 2025, doi:10.3390/ijms26073057_

Round 1
Reviewer 1 Report
Comments and Suggestions for Authors
|
Author Response
Comment 1: The introduction is limited and can be improved. It could include more information about stem cells and pluripotent stem cells.
Response 1: We believe that the manuscript will be benefited by the reviewer suggestion. The introduction has now been extended with new detailed information highlighted in yellow.
Comment 2: The introduction is limited and can be improved. It could include more information about stem cells and pluripotent stem cells.
Response 2: In attention to the comment, we included the following information related to the reviewer’s concern:
“Stem cells are endowed with the capacity for both self-renewal and specialization, making them essential for tissue maintenance, regeneration, and development. They are traditionally categorized as either embryonic stem cells (ESCs), which originate from the inner cell mass of the blastocyst, or adult stem cells, which are found in var-ious tissues, such as the bone marrow or the intestinal epithelium [21]. Although adult stem cells can differentiate into fewer cell types (often confined to a specific tissue or lineage), ESCs are considered pluripotent, meaning that they can give rise to cells of all three germ layers (ectoderm, mesoderm, and endoderm) [22].
A remarkable breakthrough in this field was the discovery of induced pluripotent stem cells (iPSCs), generated by reprogramming adult somatic cells such as fibroblasts back to a pluripotent state [23]. This process avoids several ethical and logistical hur-dles tied to ESCs and provides a patient-specific platform for research. Critically, iPSCs preserve pluripotent features, allowing scientists to guide them toward multiple line-ages—including diverse types of neurons—under controlled conditions [24]. By cre-ating iPSC-based models that integrate neuronal cells with tumor cells, researchers can conduct nuanced investigations into the ways nerves might drive or modulate cancer progression. Such systems offer a closer approximation of human biology than con-ventional cell lines do, paving the way for fresh insights into neural–tumor interactions and, potentially, novel avenues for therapeutic intervention.
A complementary strategy to iPSC-based approaches involves organoid cultures, which are three-dimensional, stem cell–derived structures that mirror key histological and functional traits of native tissues [25]. In oncology, tumor organoids capture the cellular diversity and architecture of original tumors more faithfully than traditional monolayers do while also permitting coculture with iPSC-derived neurons [26]. This setup allows the dissection of neuron–tumor interactions in an environment that better reflects in vivo conditions, thereby shedding light on the mechanisms of cancer innervation and revealing potential therapeutic targets”.
This information is highlighted in yellow between lines 63 and 89
Comment 3: Lines 45-46: Briefly discuss axonogenesis and neurogenesis.
Response 3: We further described the requested terms to provide a more detailed description of them. This information is written as follows between lines 44 and 55:
“The interactions between cancer cells and nerve cells promote neurobiological processes such as the generation of new neurons from neural stem cells [13,14]. This process is termed neurogenesis, and normally occurs either during the embryonic development and postnatal life of mammals, mainly in the subventricular and ventricular zones of the lateral ventricles, and in the dentate gyrus of the hippocampus. However, the abundance and period through which human postnatal neurogenesis persists is still on debate [15–18]. Among other mechanisms that involve increased innervation in tumors is also axonogenesis [7,8] which is the formation of an axon from a soma, and it re-quires the orchestration of multiple genes and splicing processes to undergo neuronal polarization, neurite outgrowth and axon specification so a correct axon morphogene-sis can be achieved [19,20]. Finally, neural reprogramming of sensory neurons to ad-renergic ones is also a cellular mechanism that contributes to tumor innervation [9]”.
Comment 4: The results and discussion are well described, presenting the necessary information regarding the topic addressed. However, there are acronyms throughout the text that should be explained the first time they are mentioned, for example: lines 72, 73, 76, 124, among others. Check the entire text to ensure that the acronyms are explained when first introduced
Response 4: The entire text was revised and we added acronyms where they were required. They are highlighted in yellow.
Comment 5: It was missing more representative figures; if possible, add them.
Response 5: In attention to the reviewer’s request, we added one additional figure to describe more precisely the characteristics of pluripotent stem cells (Fig. 1, line 380) and one table to synthetize the main findings of the study of the interaction between cancer cells and tumor organoids (Table 1, line 694). This information is now included in the revised version of this manuscript.
Comment 6: All the references cited are appropriate. However, the reference numbering is duplicated, and they are not formatted correctly as per the reference list format of the journal. Please correct this.
Response 6: We have addressed this issue and now there is no duplication of reference numbering.
Reviewer 2 Report
Comments and Suggestions for Authors
Given my experience in the field, the review is very interesting from a scientific point of view. To put regeneration with the use of stem cells in the foreground today is very important. In particular in this oncological context, a great perspective is evident in recognizing that preclinical testing is an essential point to study regenerative fields.
The manuscript provides a general framework under study and highlights the potential of stem cells as a solution to the question posed, with positively achievable objectives.
The bibliographic references described in this manuscript are appropriate to the study topic.
Figure no. 1 is well illustrated and represents the tumor innervation model well.
Author Response
We are grateful for the reviewer’s acknowledgement. This encourages us to continue our studies in this new exciting research field.